# Insight into Elderly ALS Patients in the Emilia Romagna Region: Epidemiological and Clinical Features of Late-Onset ALS in a Prospective, Population-Based Study

**DOI:** 10.3390/life13040942

**Published:** 2023-04-03

**Authors:** Giulia Gianferrari, Ilaria Martinelli, Cecilia Simonini, Elisabetta Zucchi, Nicola Fini, Maria Caputo, Andrea Ghezzi, Annalisa Gessani, Elena Canali, Mario Casmiro, Patrizia De Massis, Marco Curro’ Dossi, Silvia De Pasqua, Rocco Liguori, Marco Longoni, Doriana Medici, Simonetta Morresi, Alberto Patuelli, Maura Pugliatti, Mario Santangelo, Elisabetta Sette, Filippo Stragliati, Emilio Terlizzi, Veria Vacchiano, Lucia Zinno, Salvatore Ferro, Amedeo Amedei, Tommaso Filippini, Marco Vinceti, Jessica Mandrioli

**Affiliations:** 1Department of Biomedical, Metabolic and Neural Sciences, University of Modena and Reggio Emilia, 41125 Modena, Italy; 2Department of Neurosciences, Azienda Ospedaliero Universitaria di Modena, 41124 Modena, Italy; 3Clinical and Experimental Medicine Ph.D. Program, University of Modena and Reggio Emilia, 41125 Modena, Italy; 4Neuroscience Ph.D. Program, University of Modena and Reggio Emilia, 41125 Modena, Italy; 5Department of Neurology, IRCCS Arcispedale Santa Maria Nuova, 42123 Reggio Emilia, Italy; 6Department of Neurology, Faenza and Ravenna Hospital, 48100 Ravenna, Italy; 7Department of Neurology, Imola Hospital, 40026 Bologna, Italy; 8Department of Neurology, Infermi Hospital, 48018 Rimini, Italy; 9Department of Neurology, Carpi Hospital, 41012 Modena, Italy; 10Department of Biomedical and Neuromotor Sciences (DIBINEM), University of Bologna, 40126 Bologna, Italy; 11IRCCS Istituto delle Scienze Neurologiche di Bologna, UOC Clinica Neurologica, 40126 Bologna, Italy; 12Department of Neurology, Bufalini Hospital, 47521 Cesena, Italy; 13Department of Neurology, Fidenza Hospital, 43036 Parma, Italy; 14Department of Neurology, Forlì Hospital, 47121 Forlì, Italy; 15Department of Neurosciences, University of Ferrara, 44121 Ferrara, Italy; 16Department of Neurology, St. Anna Hospital, 44124 Ferrara, Italy; 17Department of General and Specialized Medicine, University Hospital of Parma, 43126 Parma, Italy; 18Department of Neurology, G. Da Saliceto Hospital, 29121 Piacenza, Italy; 19Department of Hospital Services, Emilia Romagna Regional Health Authority, 40127 Bologna, Italy; 20Department of Experimental and Clinical Medicine, University of Florence, 50134 Florence, Italy; 21Research Centre in Environmental, Genetic and Nutritional Epidemiology—CREAGEN, University of Modena and Reggio Emilia, 41125 Modena, Italy; 22School of Public Health, University of California Berkeley, Berkeley, CA 94704, USA; 23Department of Epidemiology, Boston University School of Public Health, Boston University, Boston, MA 02118, USA

**Keywords:** amyotrophic lateral sclerosis, elderly ALS, epidemiology, phenotype, prognosis, survival

## Abstract

Few studies have focused on elderly (>80 years) amyotrophic lateral sclerosis (ALS) patients, who represent a fragile subgroup generally not included in clinical trials and often neglected because they are more difficult to diagnose and manage. We analyzed the clinical and genetic features of very late-onset ALS patients through a prospective, population-based study in the Emilia Romagna Region of Italy. From 2009 to 2019, 222 (13.76%) out of 1613 patients in incident cases were over 80 years old at diagnosis, with a female predominance (F:M = 1.18). Elderly ALS patients represented 12.02% of patients before 2015 and 15.91% from 2015 onwards (*p* = 0.024). This group presented with bulbar onset in 38.29% of cases and had worse clinical conditions at diagnosis compared to younger patients, with a lower average BMI (23.12 vs. 24.57 Kg/m^2^), a higher progression rate (1.43 vs. 0.95 points/month), and a shorter length of survival (a median of 20.77 vs. 36 months). For this subgroup, genetic analyses have seldom been carried out (25% vs. 39.11%) and are generally negative. Finally, elderly patients underwent less frequent nutritional- and respiratory-supporting procedures, and multidisciplinary teams were less involved at follow-up, except for specialist palliative care. The genotypic and phenotypic features of elderly ALS patients could help identify the different environmental and genetic risk factors that determine the age at which disease onset occurs. Since multidisciplinary management can improve a patient’s prognosis, it should be more extensively applied to this fragile group of patients.

## 1. Introduction

Amyotrophic lateral sclerosis (ALS) is a rare, fatal neurodegenerative disease in which the progressive degeneration of both upper and lower motor neurons leads to muscle weakness, bulbar palsy, and, finally, death within two to five years after the onset of the first symptoms [1]. Studies on the epidemiology of ALS are numerous, and increased incidence rates of motor neuron diseases (MNDs) with aging have been reported in different western countries despite the geographic variation among these countries [2]. Some population-based studies have reported that this age-specific incidence pattern is characterized by a low incidence of ALS before 40 years of age, a progressive increase that peaks at 70–74 years of age, and a sharp decrease between the ages of 80 and 100 [3,4]. This trend has not yet been fully explained, especially for elderly patients; it could be related to misdiagnosis due to prevalent comorbidities, the presence of a faster disease that can make diagnosis challenging, or a real decrease in incidence [5]. Recently, an increase in ALS incidence among elderly women has also been described [6]. Despite the great interest in ALS epidemiology and the many open questions regarding its risk factors, few studies have explored the epidemiological and clinical features of ALS patients diagnosed at a very old age, or over 80 years (defined as oldest-old ALS by Dandaba et al. [5]). Several papers have reported an increased frequency of bulbar cases and comorbidity with dementia among the elderly [7], but other clinical features are quite unexplored. However, the population-aging effect will probably increase the number of these patients [8,9]. Furthermore, these patients represent an even more fragile subgroup of people affected by ALS, as they are neglected because they are difficult to reach and diagnose, are not followed in tertiary centers, and are not included in clinical trials [10]. Since the Emilia Romagna Region (ERR) has a super-aged society [11], with 24.33% of its population aged 65 years or over [12], we aim to investigate the demographic, clinical, and genetic features of patients with late-onset ALS compared to those with early or adult-onset ALS.

## 2. Materials and Methods

This is a prospective, population-based, epidemiological study carried out in the Emilia Romagna Region (ERR) of Northern Italy. In this region, a prospective registry (Emilia Romagna Registry for ALS–ERRALS) has been active since 2009 [13], collecting all incident ALS cases among residents of the ERR, diagnosed according to the El Escorial Criteria-Revised (EEC-R) [14].

To identify eligible patients, neurologists from 17 ALS centers collected detailed descriptions of each ALS patient including their place of residence; sex; age, both at symptom onset and diagnosis; site of onset and progression; EEC-R classification; clinical ALS phenotype, categorized as bulbar ALS, classic ALS, upper motor neuron-predominant (UMNp) ALS, flail arm and flail leg ALS, or respiratory ALS in line with the definition proposed by Chiò et al. [15]; and drug use, including Riluzole. Other collected information includes their family history and genetic analysis results. Genetic analyses were performed as per clinical practice depending on the caring physician but included at least the four genes responsible for up to 70% of familial forms of ALS [1] (SOD1, FUS, TARDBP, and the C9ORF72 expansion), as described elsewhere [16]. Depending on their family histories and previous genetic results, some patients underwent an extended and customized panel using NGS probes (Illumina Nextera Rapid Capture Custom kit, Illumina), which include the causative and susceptibility genes of ALS/FTD, hereditary motor neuron disease, and hereditary spastic paraplegia [17]. Only pathogenic mutations are reported here.

Each patient’s respiratory function, measured as their forced vital capacity (FVC), was assessed with spirometry at both diagnosis and follow-up, and their ALS Functional Rating Scale-Revised (ALSFRS-R) score assessed at each visit. Data regarding their death was also assessed. Information on the patient’s use of nutritional support, gastrostomy (PEG), and non-invasive or invasive ventilation (NIV or IV) were noted, and missing data were retrieved and confirmed through administrative data [18,19,20]. In addition, comorbidities at diagnosis were recorded, including the presence of cognitive impairment; extrapyramidal signs; cardiovascular risk factors, such as hypertension and heart diseases; diabetes mellitus; thyroid dysfunction; metabolic disorders; chronic obstructive pulmonary disease (COPD) and other respiratory disorders; gastrointestinal, urological, hematological, autoimmune, and neoplastic diseases; and psychiatric disorders [21].

We also assessed the patients’ access to diagnostic facilities (neuroimaging and electromyography (EMG)) and multidisciplinary approaches using the number of specialist evaluations they had undergone adjusted by their disease duration (number of evaluations per year), prospectively collected in the ERRALS register. Every 3–4 months, patients were given multidisciplinary follow-ups, with data on their disease progression collected according to EFNS guidelines [22]. Home monitoring of patients was performed if it was no longer possible for them to reach an ALS center for the multidisciplinary visits.

As mentioned above, disease progression was measured using ALSFRS-R scores taken at each visit and progression rates at diagnosis, which are the monthly reductions in ALSFRS-R scores given the highest score of 48 points at disease onset [23].

Weight before symptom onset, at diagnosis, and during disease progression was collected. “Weight loss at diagnosis” was used as a categorical variable, dividing patients based on whether or not they had dropped at least 1 kg of weight between the time before onset and the time of diagnosis, and as a quantitative variable, with the difference in kilograms of body weight between the times already mentioned as datapoints.

The data of the population-based registry was supplemented by cases resulting from the regional hospitals as having a discharge code of 335.2 according to the International Classification of Diseases (ICD, 9th revision) or the corresponding G12.21 code from the ICD, 10th revision, and from death certificates of residents with the same codes.

This study was approved by the ethics committee of the coordinating center (Comitato Etico Provinciale di Modena, file number 124/08, on 2 September 2008) and participating centers. For this study, we considered the incident cases that were included in the registry and diagnosed between 1 January 2009 and 31 December 2019. Patients were grouped according to two age categories: oldest-old ALS (oALS) patients, or those having an age of over 80 years at diagnosis, and patients with an age of 80 years or less at diagnosis.

Differences between means were assessed with a 2-tailed t test, or ANOVA. A comparison of categorical variables was made with a chi-square test. Ninety-five percent confidence intervals (CIs) were calculated assuming a Poisson distribution. A logistic regression analysis was performed, considering bulbar or classic phenotypes as binary response variables and age at onset (according to the following classes: <50 years, from 50 to 65 years, from 65 to 80 years, over 80 years), sex, and the interaction between age and sex as independent variables in the adjusted analysis, which was previously described [24]. Adjusted analyses for each outcome included Cox proportional hazards models for time-to-event outcomes. A Cox regression analysis was used to estimate the hazard ratio (HR) and corresponding 95% confidence interval (95% CI) for each independent variable, including gender; site of onset; phenotype; age at onset; body mass index (BMI) at diagnosis; raw value of weight loss at diagnosis (considering the weight before onset of motor symptoms as the reference); frontotemporal dementia (FTD); genetic mutations (*C9ORF72* versus other); ALSFRS-R score at diagnosis; FVC value at diagnosis; progression rate at diagnosis; diagnostic delay; use of Riluzole; family history of ALS/FTD; presence of hypertension, COPD, heart, and other respiratory diseases; autoimmune, thyroid, psychiatric, hematological, neoplastic, urologic, gastrointestinal, and metabolic disorders; and diabetes. Finally, a stepwise backward selection with a retention criterion of 0.1 was applied to the multivariate Cox regression analysis.

Statistical analyses were performed using the STATA statistical software (v15.1, StataCorp. LLC, 2017. College Station, TX, USA).

## 3. Results

### Patients’ Clinical Features

From 1 January 2009 to 31 December 2019, 222 subjects older than 80 years of age and living in the ERR received a new diagnosis of ALS, corresponding to 13.76% of all patients in the ERR diagnosed with a disease in the same period. Table 1 shows the demographic and clinical features of oALS in comparison to people who were ≤80 years old at diagnosis. Among oALS patients, the mean age at diagnosis was 83.97 ± 3.66 years, and the majority had a bulbar onset and bulbar phenotype, differing from patients who were younger (*p* < 0.001). At diagnosis, oALS patients presented with a worse clinical profile in terms of BMI (23.12 ± 3.54 vs. 24.57 ± 4.01, *p* < 0.001), weight loss (70.17% vs. 55.11%, *p* = 0.002), total ALSFRS-R scores (34.60 ± 8.82 vs. 38.97 ± 7.32, *p* < 0.001), FVC percentage (72.33 ± 27.13 vs. 86.00 ± 25.37, *p* < 0.001), and monthly decline in ALSFRS-R scores (1.43 ± 1.31 vs. 0.95 ± 1.18, *p* < 0.001).

There were only 8 patients (7 women and 1 man) diagnosed after 90 years of age; 6 out of these 8 patients had a bulbar onset and phenotype, and the mean progression rate at diagnosis was 2.69 ± 2.20 points/month.

Genetic analyses were carried out less frequently in elderly patients (25.00% vs. 39.11%, *p* < 0.001) and were negative in almost all patients. All the reported mutations are classified as pathogenic according to the ACGM classification.

Table 2 shows the distribution of phenotypes according to sex and age, where a higher frequency of the bulbar phenotype in oALS patients was detected among women.

The bulbar phenotype was associated with older ages and the female sex without interaction between age and sex. The classic phenotype, on the contrary, was inversely associated with older ages and associated with the male sex without an interaction between age and sex. The UMNp phenotype was also inversely associated with age. There were no relevant associations among other phenotypes with sex and age, according to the binary logistic regression analysis (Table 3).

Figure 1 shows the number of ALS patients in the ERR from 2009 to 2019, categorized by sex and phenotype in five-year age groups.

During the study decade, we observed an increase in the number of patients in the oldest group, with patients aged over 80 years representing 12.02% of patients before 2015 in comparison to 15.91% from 2015 onwards (*p* = 0.024).

Figure 2 displays the differences in comorbidities at the time of diagnosis between the two age groups. Comorbidities were more frequent in oALS patients than in younger patients, except for comorbidities with thyroid, psychiatric, metabolic, and respiratory diseases other than COPD.

As far as diagnostic ascertainment and the multidisciplinary approach are concerned, oALS patients underwent neuroimaging less frequently (brain MRI: 63.22% vs. 75.25%, *p* = 0–001; cervical MRI: 44.83% vs. 64.54%, *p* < 0.001), whereas access to EMG was similar between the two age groups (95.37% vs. 92.77%, *p* = 0.096) (Table 3).

The multidisciplinary team was less involved in oALS patients’ follow-ups as far as neurological, pneumological, and psychological evaluates are concerned. The oldest patients were treated less frequently with Riluzole than the others (68.79% vs. 84%, *p* < 0.001) (Table 4).

Elderly ALS patients underwent supportive procedures such as PEG, NIV, or IV earlier in the course of the disease but less frequently than other patients (Table 5).

The elderly had a faster disease progression and a shorter survival length compared to younger ALS patients (Figure 3); the median tracheostomy-free survival length was 36 months (95% CI 33.57–37.43) in ALS patients who were ≤80 years old at diagnosis and 20.77 months (95% CI 18–23.93) in oALS patients (HR 1.87, 95% CI 1.61–2.17, *p* < 0.001)

In oALS patients, the univariate analysis of the length of tracheostomy-free survival demonstrated that negative prognostic factors, included in the multivariable model, were weight loss at diagnosis (kg) (HR = 1.54, 95% CI: 0.98–2.42, *p* = 0.060), progression rate at diagnosis (HR = 1.53, 95% CI: 1.38–1.70, *p* < 0.001), age at onset (years) (HR = 1.04, 95% CI: 1.01–1.07, *p* = 0.007), and the presence of cardiovascular diseases (HR = 1.44, 95% CI: 1.02–2.03, *p* = 0.036), psychiatric diseases (HR = 1.84, 95% CI: 0.90–3.79, *p* = 0.093), or hematological diseases (HR = 0.53, 95% CI: 0.26–1.09, *p* = 0.083), whereas higher BMIs (HR = 0.92, 95% CI: 0.87–0.97, *p* = 0.005) and ALSFRS-R scores at diagnosis (1 point) (HR = 0.97, 95% CI: 0.96–0.99, *p* = 0.010) and longer diagnostic delays (months) (HR = 0.97, 95% CI: 0.94–0.97, *p* < 0.001) were associated with longer survivals (Table 6).

Independent prognostic factors for the length of tracheostomy-free survival in the multivariate analysis were ALSFRS-R score at diagnosis (1 point) (HR = 0.97, 95% CI: 0.95–0.98, *p* < 0.001), age at onset (years) (HR = 1.06, 95% CI: 1.00–1.11, *p* = 0.039), and diagnostic delay (months) (HR = 0.97, 95% CI: 0.95–0.98, *p* < 0.001) (Figure 4).

## 4. Discussion

From 1 January 2009 to 31 December 2019, 14% of all new ALS cases diagnosed in the ERR were in subjects older than 80 years of age, a similar statistic to what was reported in the FRALim register [9]. These patients appear to have specific clinical features that differ from the remaining ALS population; elderly ALS patients are more likely to be female and show a higher prevalence of bulbar symptoms at onset and of the bulbar phenotype. We can confirm previous findings from the Scottish MND register [25] and the Piemonte and Valle d’Aosta register for ALS (PARALS) populations [24] with the evidence that age and sex also influence the phenotypes of our elderly study population. We could not find any evidence of an interaction among age and sex, probably as a result of the relatively small sample size, especially among the elderly. A relationship between very late-onset ALS and predominantly bulbar presentations was also already found in cohort studies [26].

This study found that, with increasing age, patients’ clinical statuses at diagnosis were worse in terms of BMI, ALSFRS-R, and FVC scores [5,27], and they had significantly higher progression rates and reduced lengths of survival compared to other ALS patients [27].

In our regional cohort, approximately 15% of patients reported a family history of ALS/dementia (fALS), among both the elderly and patients who were ≤80 years at diagnosis. Although there were no significant differences between age groups as far as family history is concerned, genetic testing was carried out less frequently among elderly patients, and only 5.45% of them showed a mutation in the genes related to ALS, namely *SOD1* and *FUS*. These results are quite distinct from those in previous studies, which reported that, among patients carrying mutations in genes related to ALS, the elderly were the ones carrying the *C9ORF72* expansion [28]. This discrepancy could be attributable in part to the low rate of genetic testing among the elderly in our population. The presence of *FUS* mutations in oALS patients is atypical because it represents the most common genetic defect in early onset ALS (<40 years) [28,29,30]. Some of the data in the literature, however, show that patients with an *FUS* mutation exhibit considerable variation in phenotypes, with some having an early onset and rapid disease progression and others having a later age of onset and slower progression [31]. This variability in the phenotypes of FUS-related ALS could result from the effects exerted by different missense and truncating mutations [32], or there could be influences from epigenetics or environmental factors on disease phenotype in patients carrying mutations. The lower frequency of pathogenic mutations detected among oALS patients could suggest that pathogenic mutations lead to an increased risk of early disease onset, according to the multistep hypothesis of ALS pathogenesis [33,34]. On the other hand, it cannot be ruled out that patients with a disease onset beyond 80 years of age carry mutations in unknown susceptibility genes.

Nevertheless, the fact that genetic analyses were performed based on clinical judgment may bias these results, and we acknowledge that this as a limitation of this study. The reason behind the low propensity for genetic testing could be the current clinical practice of performing genetic tests only on individuals with a positive family history or young people with sporadic diseases, although it is likely that genetic mutations are present in patients with apparently sporadic ALS at all ages [35]. With the use of precision treatments rising (e.g., antisense oligonucleotides), it is essential to identify all ALS patients carrying genetic mutations. Age at onset and family history should not be an obstacle to genetic testing. On the other hand, it must be considered that variants of unknown significance and uncertainties related to incomplete penetrance may require expert genetic and psychological counseling [36,37].

During this study, we observed an increase in the proportion of subjects over 80 years of age, probably reflecting the population aging [38], the accuracy of detection, and improved surveillance and reporting in the Emilia Romagna Region [6,13]. On the other hand, this increasing number of elderly patients might be due to not only better case ascertainment but also a combination of environmental, genetic, and individual factors that can contribute to the heterogeneity of disease presentation, including age at onset.

Our study highlights a generally poor prognosis for elderly ALS patients that could be related to the higher prevalence of comorbidities, such as cardiovascular diseases (e.g., hypertension, congestive heart failure, atrial fibrillation, and coronary disease) and hematological disorders, that were previously associated with a faster disease progression [21,39].

Comorbidities, an atypical clinical onset, and a faster disease progression can also complicate the evaluation and management of ALS by making the diagnosis more challenging [3,4,40,41,42]. Cognitive impairment, which is more prevalent with age, may limit an individual’s ability to provide an accurate history of symptoms. Moreover, access to neurological specialists may be limited for older adults who are institutionalized in nursing homes or long-term care facilities, or those with low health-care education. Elderly people come to the attention of neurological centers with greater difficulty or later in time [40,41], with a higher risk of being underdiagnosed or misdiagnosed. This may explain the underestimation of oALS patients in previous studies [3,4,43]. Accordingly, in our patients’ cohort, oALS patients underwent some diagnostic investigations, such as neuroimaging studies, less frequently. However, diagnostic delay in oALS patients, in the context of a faster disease, was not significantly different from other ALS patients [8,24].

This study points out that only 69% of the oldest-old patients had been prescribed Riluzole, the only disease-modifying therapy currently approved in Italy. This could be related to the patients’ worse clinical statuses at the time of diagnosis and the higher frequency of FTD or cognitive impairment among older ALS patients (10,98% vs. 7.87%), although this difference is not statistically significant. This observation could represent a limitation to healthcare access, given the aging of the Western population, that needs to be accounted for using population-based studies [9]. Multidisciplinary teams were also less involved in the management of oALS patients (especially as far as neurological, pneumological, and psychological evaluations are concerned), partly as a result of the shorter length of ALS survival in these patients. This is probably also due to the more frequent denial of support procedures by oALS patients, which is more likely to be a patient’s choice than ageism. Although the involvement of specialist palliative care is of undoubtable value for every patient [44], multidisciplinary management in close cooperation with general practitioners, home carers, and a dedicated healthcare network should also be recommended for the proper assessment of all patients to minimize morbidity and maximize their quality of life [44,45].

Together with differences in healthcare management, more severe frontotemporal cortical thinning [46], and worse clinical statuses at disease onset and diagnosis (such as lower BMI, lower ALSFRS-R score, and higher progression rate) [47,48,49,50] in elderly patients, the harmful effects of aging on known prognostic factors [8] may contribute to the worse prognoses of these patients.

On the contrary, the bulbar-onset form, which is a well-known negative predictor of survival in the general ALS population [15,47], was not significantly associated with poor prognoses in oALS patients. This could be explained by the fact that other disease factors may have a stronger impact on survival in this age group. Consistently, the multivariate analysis of length of survival showed that an older age at onset, ALSFRS-R scores at diagnosis, and shorter diagnostic delays are independently associated with poor survival outcomes among the elderly [51,52].

## 5. Conclusions

Our study shows that elderly ALS patients are represented more frequently by women and tend to have a bulbar onset, a more rapid disease course, and shorter length of survival. Furthermore, we documented the limited application of treatments and procedures in this group. The main strengths of our study are the use of a population-based registry for data collection, involving an extensive network of ALS neurologists in the ERR, the length of follow-up, and the high number of cases described, reflecting the high accuracy of case ascertainment. Some limitations should be recognized in the presence of missing data on clinical phenotypes, cognitive/behavioral impairment details, and disease progression rates.

To improve the diagnosis, prognosis, and quality of life in patients in this extremely fragile group, diagnostic tests and multidisciplinary management should be more extensively applied and aimed at minimizing diagnostic latency.

## Figures and Tables

**Figure 1 life-13-00942-f001:**
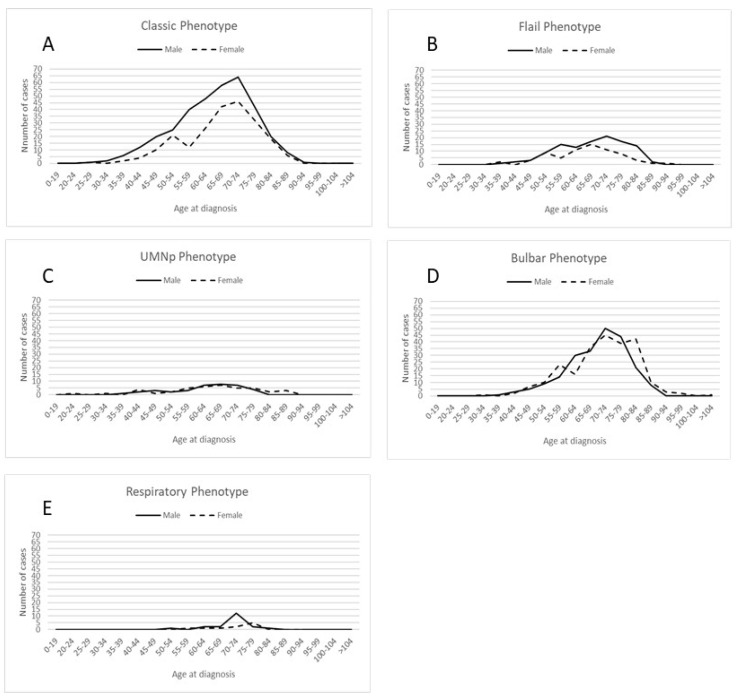
Distribution of cases by sex and 5–year age groups across different disease phenotypes: (**A**) classic phenotype; (**B**) flail phenotype, including both flail arm and flail leg phenotypes; (**C**) Upper Motor Neuron predominant (UMN-p) phenotype; (**D**) bulbar phenotype; (**E**) respiratory phenotype.

**Figure 2 life-13-00942-f002:**
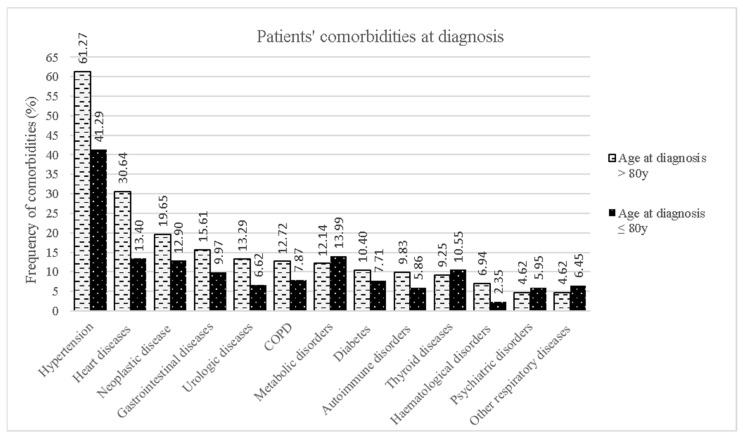
Distribution of comorbidities at diagnosis for the two age groups.

**Figure 3 life-13-00942-f003:**
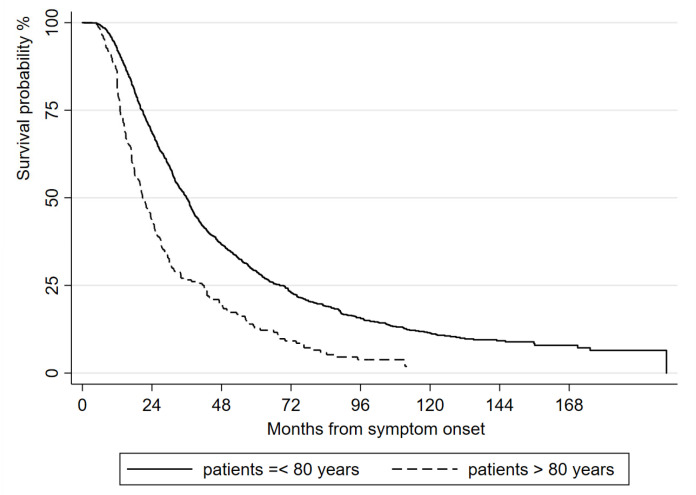
Kaplan-Meier analysis of length of tracheostomy-free survival from symptom onset, comparing oALS and other ALS patients.

**Figure 4 life-13-00942-f004:**
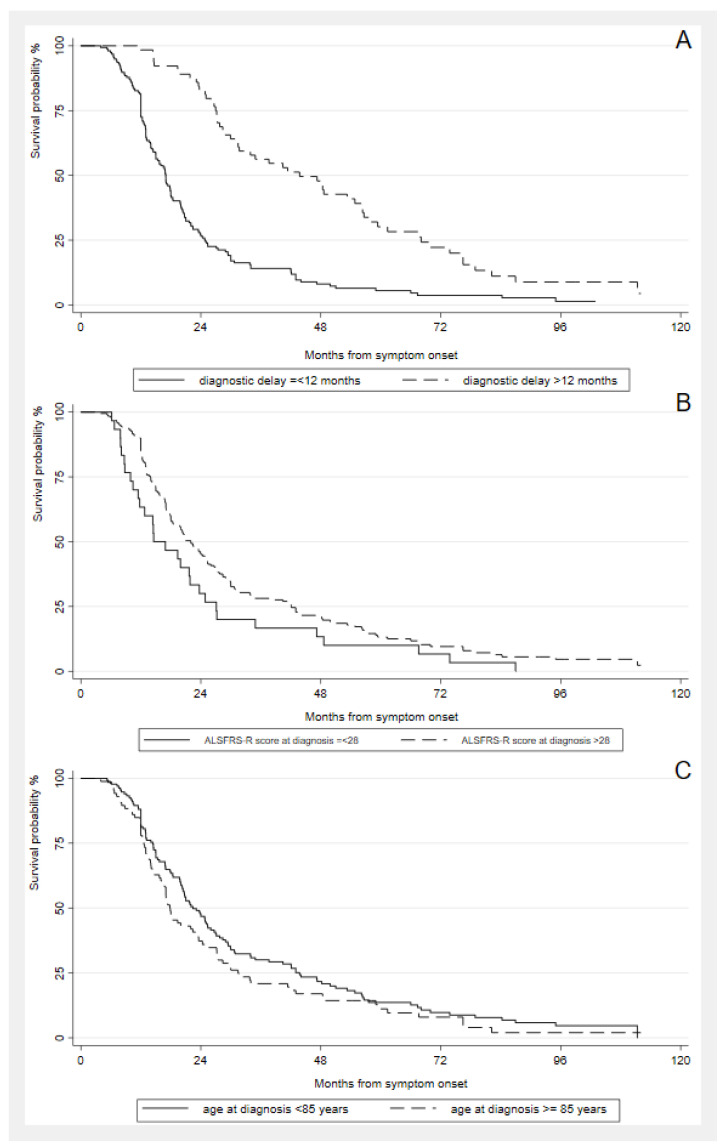
Kaplan-Meier analysis of length of tracheostomy-free survival from symptom onset in oALS patients by (**A**) diagnostic delay (≤ or >12 months); (**B**) ALSFRS-R score at diagnosis (≤ or > 28/48); (**C**) age classes (< or ≥ 85 years).

**Table 1 life-13-00942-t001:** Demographic and clinical features of patients according to age group at diagnosis.

	Age at Diagnosis ≤80 years		Age at Diagnosis >80 years		Total		*p* Value
	N or Mean	% or SD	N or Mean	% or SD	N or Mean	% or SD	
Male sex	783	56.29	102	45.94	885	54.87	0.004
Mean age at onset, years	64.44	10.13	82.83	3.76	67.01	11.42	<0.001
Mean age at diagnosis, years	65.53	10.12	83.97	3.66	68.06	11.42	<0.001
Mean diagnostic delay, months	13.66	13.73	13.71	11.56	13.67	13.44	0.954
Site of onset °							<0.001
Bulbar	373	32.58	85	51.52	458	34.96	<0.001
Spinal UL	381	33.28	33	20.00	414	31.60	<0.001
Spinal LL	361	31.53	47	28.48	408	31.15	0.128
Respiratory	30	2.62	0	0.00	30	2.29	0.012
Phenotype °							<0.001
Bulbar	369	32.09	86	52.43	455	34.63	<0.001
Classic	515	44.78	52	31.71	567	43.15	<0.001
Flail Arm	58	5.04	5	3.05	63	4.79	0.171
Flail Leg	105	9.13	15	9.15	120	9.13	0.676
UMNp	74	6.43	5	3.05	79	6.01	0.049
Respiratory	29	2.52	1	0.61	30	2.28	0.094
FTD °	94	7.87	19	10.98	113	8.27	0.165
Family history of ALS/FTD °	180	15.07	21	12.14	201	14.70	0.308
BMI at diagnosis °, Kg/m^2^	24.57	4.01	23.12	3.54	24.41	3.99	<0.001
Weight loss at diagnosis °, number	507	55.11	80	70.17	587	56.77	0.002
Weight loss at diagnosis, kg	3.92	6.02	5.14	6.49	4.06	6.09	0.043
Progression rate at diagnosis *, points/month	0.95	1.18	1.43	1.31	1.01	1.21	<0.001
Mean ALSFRS-R score at diagnosis °, points	38.97	7.32	34.60	8.82	38.44	7.64	<0.001
Mean FVC value at diagnosis °, %	86.00	25.37	72.33	27.13	84.63	25.85	<0.001
Genetic analysis °							0.111
no mutations	471	86.58	52	94.55	523	87.31	0.091
C9ORF72 expansion	39	7.17	0	0.00	39	6.51	0.040
SOD1 mutations	17	3.13	1	1.82	18	3.01	0.589
FUS mutations	5	0.92	2	3.64	7	1.17	0.074
TARDBP mutations	4	0.74	0	0.00	4	0.67	0.523
Other genes involvement	8	1.47	0	0.00	8	1.34	0.365
Death	1046	76.68	202	91.40	1248	78.73	<0.001
Total	1391	86.24	222	13.76	1613	100	

SD: standard deviation; spinal UL: spinal upper limb; spinal LL: spinal lower limb; UMNp: upper motor neuron predominant; FVC: forced vital capacity; FTD: frontotemporal dementia. * Progression rate at diagnosis was calculated as the monthly decline in ALSFRS-R scores, assuming a total score of 48 at onset; it was calculated for 1162 total patients. ° Data not available for all patients; FVC% value was listed for 600 subjects; family history and FTD data were available for 1367 patients, weight loss for 1034 cases, and BMI for 1114 cases. Onset type was available for 1310 cases, and the phenotype for 1314. Genetic tests were performed on 599 patients.

**Table 2 life-13-00942-t002:** Distribution of phenotypes according to sex in oALS and younger ALS patients.

	Male Patients					Female Patients				
	Age at Diagnosis ≤80 years		Age at Diagnosis >80 years		*p* Value	Age at Diagnosis ≤80 years		Age at Diagnosis >80 years		*p* Value
	N	%	N	%		N	%	N	%	
Phenotype										
Bulbar	189	28.51	29	39.73	0.344	**180**	**36.96**	**57**	**62.64**	**<0.001**
Classic	**319**	**48.11**	**28**	**38.36**	**0.010**	**196**	**40.24**	**24**	**26.37**	**0.008**
Flail arm	35	5.28	4	5.48	0.819	23	4.72	1	1.10	0.098
Flail leg	64	9.65	11	15.07	0.373	41	8.42	4	4.40	0.156
UMNp	**37**	**5.58**	**0**	**0.00**	**0.025**	37	7.60	5	5.49	0.410
Respiratory	19	2.87	1	1.37	0.355	10	2.05	0	0.00	0.157
Total	663	100.00	73	100.00		487	100.00	91	100.00	

UMNp: upper motor neuron predominant.

**Table 3 life-13-00942-t003:** Correlation of motor phenotypes with age and sex: binary logistic regression analysis.

Motor Phenotype	Factor	Level	OR (95% CI)	*p* Value
Bulbar (*n* = 455)	Sex, female		1.45 (1.13–1.87)	0.004
	Age	<50 years	1	
		50–65 years	1.50 (0.082–2.76)	0.189
		65–80 years	2.29 (1.37–3.83)	0.002
		>80 years	3.14 (1.79–5.51)	<0.001
	Age x sex		1.08 (0.65–1.81)	0.758
Classic (*n* = 567)	Sex, male		1.34 (1.05–1.72)	0.019
	Age	<50 years	1	
		50–65 years	0.61 (0.37–1.01)	0.054
		65–80 years	0.58 (0.39–0.85)	0.006
		>80 years	0.35 (0.22–0.56)	<0.001
	Age x sex		1.25 (0.79–2.00)	0.341
Flail arm (*n* = 63)	Sex, male		1.56 (0.80–3.04)	0.189
	Age	<50 years	1	
		50–65 years	4.60 (0.92–23.02)	0.063
		65–80 years	2.49 (0.59–10.53)	0.216
		>80 years	1.45 (0.28–0.63)	0.659
	Age x sex		0.65 (0.22–1.88)	0.425
Flail leg (*n* = 120)	Sex, male		1.54 (0.97–2.46)	0.068
	Age	<50 years	1	
		50–65 years	1.47 (0.58–3.70)	0.414
		65–80 years	0.95 (0.46–1.99)	0.903
		>80 years	1.02 (0.43–2.40)	0.960
	Age x sex		0.72 (0.32–1.65)	0.443
UMNp (*n* = 79)	Sex, male		0.67 (0.38–1.16)	0.155
	Age	<50 years	1	
		50–65 years	0.49 (0.21–1.179	0.110
		65–80 years	0.36 (0.19–0.71)	0.003
		>80 years	0.17 (0.06–0.51)	0.001
	Age x sex		0.97 (0.36–2.59)	0.955
Respiratory (*n* = 30)	Sex, male		1.84 (0.78–4.31)	0.161
	Age	<50 years	1	
		50–65 years	3.39 (0.29–39.91)	0.331
		65–80 years	6.40 (0.86–47.64)	0.070
		>80 years	n.o.	
	Age x sex		0.59 (0.08–4.32)	0.604

OR: odds ratio; UMNp: upper motor neuron predominant; n.o.: no observations.

**Table 4 life-13-00942-t004:** Diagnostic and clinical approaches to oALS patients compared to other ALS patients.

	Age at Diagnosis ≤80 years		Age at Diagnosis >80 years		Total		*p* Value
	N or Mean	% or SD	N or Mean	% or SD	N or Mean	% or SD	
Use of Riluzole *	1003	84.00	119	68.79	1122	82.08	<0.001
Diagnostic procedures °							
EMG	1129	92.77	165	95.37	1294	93.09	0.096
Brain MRI	921	75.25	110	63.22	1031	73.75	0.001
Cervical MRI	790	64.54	78	44.83	868	62.09	<0.001
Lumbosacral MRI	405	33.09	43	24.71	448	32.04	0.027
Multidisciplinary evaluation							
Neurological examinations, n/year	2.25	1.87	1.93	1.38	2.21	1.82	0.035
Pneumological evaluations, n/year	1.44	1.31	1.20	1.02	1.41	1.28	0.025
Medical rehabilitation assessments, n/year	1.00	1.02	0.90	0.91	0.98	1.01	0.235
Speech therapist assessments, n/year	0.65	0.85	0.64	0.81	0.65	0.81	0.900
Nutritional assessments, n/year	0.99	1.19	0.95	1.05	0.98	1.17	0.729
Psychological assistance, n/year	0.65	0.92	0.22	0.71	0.62	0.90	0.002
Palliative care evaluations, n/year	0.22	0.67	0.18	0.52	0.22	0.65	0.432
Medical and nursing home care, n/year	0.42	0.96	0.30	0.59	0.41	0.92	0.113
Total	1391	86.24	222	13.76	1613	100	

* Data available for 1367 patients; ° data available for 1398 patients, except for EMG (1390 cases).

**Table 5 life-13-00942-t005:** Nutritional and respiratory support procedures and patient features at the time according to two age groups.

	Age at Diagnosis ≤80 years		Age at Diagnosis >80 years		Total		*p* Value
	N or Mean	% or SD	N or Mean	% or SD	N or Mean	% or SD	
Nutritional Support (PEG)							
PEG	371	31.05	36	20.81	407	29.75	0.006
time onset-PEG, months	26.24	16.09	19.41	14.18	25.65	16.03	0.017
Non-invasive ventilation (NIV)							
NIV	454	38.06	50	28.90	504	36.90	0.020
time onset-NIV, months	27.57	23.44	21.87	15.68	27.01	22.84	0.104
Invasive ventilation (IV)							
IV	216	18.09	12	6.94	228	16.68	<0.001
time onset-IV, months	29.80	19.84	16.18	11.58	29.13	19.72	0.025

Data available for 1367 patients.

**Table 6 life-13-00942-t006:** Univariate Cox regression analysis of survival in oldest-old ALS patients.

Variable	HR (95% CI)	*p* Value
Gender		
Female sex	1	
Male sex	0.99 (0.75–1.31)	0.956
Site of onset	1.05 (0.95–1.15)	0.330
Bulbar		
Spinal UL		
Spinal LL		
Respiratory		
Phenotype	1.04 (0.98–1.10)	0.231
Bulbar		
Classic		
Flail Arm		
Flail Leg		
UMN-p		
Respiratory		
FTD	1.44 (0.89–2.34)	0.136
Genetic analysis (C9ORF72 expansion vs. other mutations)	0.73 (0.40–1.32)	0.296
Weight loss at diagnosis, kg	1.54 (0.98–2.43)	0.060
BMI at diagnosis, kg/m^2^	0.92 (0.87–0.97)	0.005
ALSFRS-R score at diagnosis, points	0.98 (0.96–0.99)	0.010
Progression rate at diagnosis, points/months	1.53 (1.38–1.70)	<0.001
Age at onset, years	1.04 (1.01–1.07)	0.007
Diagnostic delay, months	0.95 (0.94–0.97)	<0.001
FVC value at diagnosis, %	0.99 (0.98–1.00)	0.258
Use of Riluzole	0.81 (0.58–1.14)	0.239
Family history of ALS/FTD	0.95 (0.58–1.56)	0.849
Hypertension	0.87 (0.63–1.21)	0.425
COPD	1.20 (0.75–1.92)	0.451
Other respiratory diseases	0.70 (0.31–1.60)	0.402
Heart diseases	1.44 (1.02–2.03)	0.036
Autoimmune disorders	0.98 (0.57–1.70)	0.950
Diabetes	0.76 (0.46–1.27)	0.293
Thyroid diseases	0.89 (0.51–1.54)	0.678
Psychiatric disorders	1.84 (0.90–3.79)	0.093
Hematological disorders	0.53 (0.26–1.09)	0.083
Neoplastic disease	1.11 (0.76–1.64)	0.588
Urologic diseases	0.83 (0.52–1.32)	0.430
Gastrointestinal diseases	0.94 (0.61–1.46)	0.793
Metabolic disorders	0.97 (0.59–1.58)	0.891

Spinal UL: spinal upper limb; spinal LL: spinal lower limb; UMN-p: upper motor neuron predominant; FTD: frontotemporal dementia. BMI: body mass index; ALSFRS-R: ALS Functional Rating Scale-Revised; progression rate at diagnosis was calculated as the monthly decline in ALSFRS-R scores, assuming a total score of 48 at onset; FVC: forced vital capacity; family history of ALS/FTD: family history of amyotrophic lateral sclerosis/frontotemporal dementia; COPD: chronic obstructive pulmonary disease; HR: hazard ratio; CI: confidence interval.

## Data Availability

Data are available from the authors upon reasonable request.

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
