# Peer review of "Insight into Elderly ALS Patients in the Emilia Romagna Region: Epidemiological and Clinical Features of Late-Onset ALS in a Prospective, Population-Based Study"

_life, 2023, doi:10.3390/life13040942_

Round 1

Reviewer 1 Report

Gianferrari et al describe the characteristics of old-onset ALS patients in a population-based cohorts. The findings, although already described, are of interest given the increasingly older population and the paper is overall well-written. I have, however, some suggestions for improvement:

-          Abstract:

o   Please provide data on BMI and progression rate, when describing differences between oALS vs younger ALS patients.

o   “They increased by 18% from 2009-2014 to 2015-2019”à It is unclear what this 18% means. I found it better explained in the text: oALS patients represented 12.02% of patients before 2015 in comparison to 15.91% from 2015 onwards (p=0.024), what means that its frequency increased an 18% in 5 years.

-          Methods:

o   Please detail how many ALS centres are found in the ERR region.

o   “Weight in health status”à Please, name it weight before motor symptoms onset.

o   weight loss is reported as a categorical variable (yes/no). It would be better to report it as a quantitative variable (in kg). If this data is not available, authors should explain what especifically is considered weight loss (>0kg, >1kg??).

o   Please, specify which were the criteria for genetic analysis and which analysis were perfomed in each patient (C9 only? Gene panel??...). Does this varies among the centres? Because this is key to interpret the genetic results. If the criteria vary among centres, please add this as a limitation and be cautious when interpreting the results of genetic analysis.

o   Please, explain how the multidisciplinary evaluation (neurological examination…) was calculated. It should be adjusted by the disease duration. Otherwise, it could just be the result of older patients having shorter survivals

o   Logistic regression is performed, but this is not mentioned in the results. Moreover, the variables included in each multivariable analysis should be commented in the methods and the results should be provided, at least as supplementary analysis (not just commented, see below).

-          Results:

o   It would be interesting to know how many patients older than 90 yo were found.

o   Also, how many patients (>80 and <80) were visited in regional centres vs in ALS units? Do you have this information?

o   “weight loss (55.11%, 70.17%, p=0.002)” à Please change the order: weight loss (70.17%, 55.11%, p=0.002).

o   In Table 1 and 2, please, indicate the sample size for those variables in which data are missing, and recalculate the percentages after excluding patients with no data available (to sum up 100%), so that percentages are comparable between groups. Also, please, indicate units for BMI, weight loss, progression rate, FVC...

o   In Table 2, when comparing the total number of subjects, no p-value is needed.

o   How do authors define respiratory and bulbar phenotype and why are these numbers different to respiratory and bulbar onset?

o   Please, provide more details (nomenclature) on the mutations found in oALS patients and their ACGM classification

o   “Correlation of bulbar phenotypes with age and sex by”à Please, delete this. Is unnecessary and leads to misunderstanding (regression measures association, no correlation).

o   Please, provide the full results of the regression analysis that are commented in the text, at least as supplementary material. Moreover, it is surprising that no interaction between age and sex was found in bulbar-onset patients, since in the figure it seems clears that women incidence of bulbar onset peaks about 10 years later than men. This does not happen in classical ALS. Please, comment on this in the discussion. Do you think there is such effect or not?

o   Percentages on cervical MRI are wrong. Please, correct them.

o   Was bulbar onset not associated with shorter survival in the cox analysis? Please, provide tables and comment on this in the discussion because this is atypical, as authors acknowledge in the discussion (bulbar-onset form, which is a well-known neg-316 ative predictor of survival [15,39]).

-          Discussion:

o   Only 5% of the studied oALS patients vs 15% of the studied adult-onset ALS patients were found to carry a mutation. Does this suggests that older patients carry more frequently unknown/undiscovered mutations?

o   The finding of FUS mutations in oALS is atypical because they are usually associated to earlier onset. Please comment on that.

o   Authors found that oALS live shorter and they attributed it to comorbidities, disease progression rate, etc… However these patients also received less frequently life-prolonging treatments (riluzol, NIV, PEG…). Please, comment on this. Moreover, why do authors think they received less frequently these treatments? Do they think this is ageism or are other reasons (patients preferences, etc…). Please comment.

-          Some typos:

o   Abstract, row 58 (because more difficult)à because they are more difficult.

o   Results, row 191 (less associated)à inversely associated

o   Discussion, row 281 (With incoming) à with the upcoming of

o   Discussion, row 287 (we observed an increase in subjects) à we observed an increase in the proportion of subjects

o   Discussion, row 306 (less frequently to some diagnostic)à please, delete “to”

Reviewer 2 Report

Dear authors

This is a very well-documented and well-documented article on late onset ALS, and covers a significant number of different clinical aspects of the disease. Statistical analysis is also well done. However, it is better to pay more attention to some points.

Key words:

1.    There are many keywords and some of them are unnecessary.

2.    It is recommended to choose keywords from MeSH words to facilitate article retrieval during searching.

Introduction:

1.    In many similar articles, “late onset ALS” after the age of 70 and “very late onset ALS” after the age of 80 have been mentioned. It is better to mention this in the introduction, and state what exactly is meant by oldest-old ALS in this study?

Methods:

1.    How much weight difference is considered weight loss?

Reviewer 3 Report

Giulia Gianferrari et.al has put a nice clinical population based study. The paper is fit for publication.

Minor: The authors can put the table in a nicer way representing the mean and standard deviation in separate columns for readers to better understand the data.
